# Use of Generalized Weighted Quantile Sum Regressions of Tumor Necrosis Factor Alpha and Kidney Function to Explore Joint Effects of Multiple Metals in Blood

**DOI:** 10.3390/ijerph19127399

**Published:** 2022-06-16

**Authors:** Kuei-Hau Luo, Hung-Pin Tu, Cheng-Hong Yang, Chen-Cheng Yang, Tzu-Hua Chen, Hung-Yi Chuang

**Affiliations:** 1Graduate Institute of Medicine, College of Medicine, Kaohsiung Medicine University, Kaohsiung City 807, Taiwan; u107800007@kmu.edu.tw (K.-H.L.); u106800001@kmu.edu.tw (C.-C.Y.); 2Department of Public Health and Environmental Medicine, College of Medicine, Kaohsiung Medicine University, Kaohsiung City 807, Taiwan; p915013@kmu.edu.tw; 3Department of Electronic Engineering, National Kaohsiung University of Science and Technology, Kaohsiung City 807, Taiwan; chyang@nkust.edu.tw; 4Department of Occupational and Environmental Medicine, Kaohsiung Medicine University Hospital, Kaohsiung Medicine University, Kaohsiung City 807, Taiwan; 5Department of Family Medicine, Kaohsiung Municipal Ta-Tung Hospital, Kaohsiung 80145, Taiwan; 980264@kmuh.org.tw; 6Ph.D. Program in Environmental and Occupational Medicine, and Research Center for Environmental Medicine, College of Medicine, Kaohsiung Medical University, Kaohsiung 807, Taiwan

**Keywords:** TNF-α, arsenic (As), cadmium (Cd), lead (Pb), cobalt (Co), copper (Cu), selenium (Se), zinc (Zn), weighted quantile sum (WQS) regression, generalized additive model (GAM)

## Abstract

Exposure to heavy metals could lead to adverse health effects by oxidative reactions or inflammation. Some essential elements are known as reactors of anti-inflammatory enzymes or coenzymes. The relationship between tumor necrosis factor alpha (TNF-α) and heavy metal exposures was reported. However, the interaction between toxic metals and essential elements in the inflammatory response remains unclear. This study aimed to explore the association between arsenic (As), cadmium (Cd), lead (Pb), cobalt (Co), copper (Cu), selenium (Se), and zinc (Zn) in blood and TNF-α as well as kidney function. We enrolled 421 workers and measured the levels of these seven metals/metalloids and TNF-α in blood; kidney function was calculated by CKD-EPI equation. We applied weighted quantile sum (WQS) regression and group WQS regression to assess the effects of metal/metalloid mixtures to TNF-α and kidney function. We also approached the relationship between metals/metalloids and TNF-α by generalized additive models (GAM). The relationship of the exposure–response curve between Pb level and TNF-α in serum was found significantly non-linear after adjusting covariates (*p* < 0.001). Within the multiple-metal model, Pb, As, and Zn were associated with increased TNF-α levels with effects dedicated to the mixture of 50%, 31%, and 15%, respectively. Grouped WQS revealed that the essential metal group showed a significantly negative association with TNF-α and kidney function. The toxic metal group found significantly positive associations with TNF-α, serum creatinine, and WBC but not for eGFR. These results suggested Pb, As, Zn, Se, and mixtures may act on TNF-α even through interactive mechanisms. Our findings offer insights into what primary components of metal mixtures affect inflammation and kidney function during co-exposure to metals; however, the mechanisms still need further research.

## 1. Introduction

Heavy metals, for instance, arsenic (As), cadmium (Cd), and lead (Pb), are widely utilized in industry [1]. Based on the International Agency for Research on Cancer, Pb was grouped as a metal that is probably carcinogenic to humans (group 2), and As and Cd were classified as carcinogenic substances (group 1) [2]. Unfortunately, the dramatic global increase in urbanization and industrialization has increased the risk of exposure to such heavy metals. Multiple heavy metal exposure has become a serious public health issue [3,4,5]. These toxic metals or non-essential metals lead to adverse health effects, such as cardiovascular, kidney, liver, nervous system, and skin damage [6,7,8]. An increasing number of studies have shown that the crucial effects underlying diseases associated with heavy metal exposure are inflammation and immune responses [9,10]. These health effect studies were conducted with heavy metal exposure in a variety of populations. As we know, the toxicity of As, Cd, and Pb are closely related to kidney function and inflammation.

Nevertheless, several metal/metalloids are essential elements that play key roles in metabolism and antioxidant reactions [11]. Cobalt (Co), copper (Cu), selenium (Se), and zinc (Zn) are reported to be essential metals [12]. Understanding the interactions between multiple metals during exposures of workers’ health is essential, especially in complex occupational scenarios with multiple potential health risks.

Tumor necrosis factor-alpha (TNF-α) is primarily released from macrophages [13]. TNF-α has been noted for its pro-inflammatory effect [14] and its close relation to numerous diseases [15,16,17]. There have been several studies that have investigated the relationship between heavy metals and TNF-α. Studies using in vitro models with Cd, Pb, and As showed that these metals/metalloids induced pro-inflammatory states [18,19,20]. Research has found that levels of TNF-α are associated with Pb toxicity [21,22]. Long-term As exposure also leads to a high level of inflammatory markers [23,24], and blood cadmium levels have also been associated with TNF-α levels [25]. On the other hand, essential elements play key roles in the related molecular mechanisms of metabolism, enzyme activity, and anti-inflammatory cytokines. Akyuva et al. reported that Se decreased the inflammation of microglia [26]. Another study evaluated the restoration of plasma Zn concentrations, leading to a decrease in oxidative stress in hemodialysis patients [27]. Liu et al. found that a deficiency in Cu, which increased TNF-α overexpression, was associated with lung inflammation in mice [28]. However, multiple metal substances exist in the environment concurrently and potentially correlate with the health effects of each element, therefore, the health effects of metal co-exposure should receive more attention.

Occupational exposure to heavy metals during manufacturing processes is well documented in many industries [29,30]. Workers are exposed to multiple metals through inhalation, intake, and dermal contact. The health effects of heavy metals are related to both long- and short-term exposure to these heavy metals. Animal studies have found oxidative damage from As, Cd, and Pb [31,32,33]. Some studies have proved the interaction between essential metals and toxic metals [34,35]; however, the mechanisms of multiple metal exposure are unclear. Identifying the inflammation of metal mixtures is troublesome due to the independent, antagonistic, or additive interactions between toxic metals and essential elements. Weighted quantile sum (WQS) regression is more fitted to approaching environmental risk factors than the conventional regression model [36]. The WQS method analyzes high-dimensional datasets, such as those for metal/metalloid mixtures, through a weighted index that estimates the mixed effect of all predictor variables on the outcome. Through the weighted index, we can explore the combined effects of multiple metals/metalloids on TNF-α with co-adjustments for the overall relevant covariates.

In this study, we explored the associations among seven serum biomarkers for metals, including white blood cells, serum creatinine, TNF-α, and estimated glomerular filtration rate (eGFR). In order to evaluate the effect of single-metal and metal mixture exposures on outcomes, we applied four models: (i) the linear regression model, (ii) the generalized additive model (GAM), (iii) the generalized weighted quantile sum (WQS) regression [37], and (iv) the grouped weighted quantile sum regression (GWQS). Our three aims were, first, to survey the associations between single metals and TNF-α levels and renal function. Then, we assessed the relationship between metals and TNF-α levels. Finally, we determined the interrelation between the metal mixtures and TNF-α levels and kidney function.

## 2. Materials and Methods

### 2.1. Study Population

The participants for this research were 462 workers enrolled in annual health examinations between 2009 and 2010 in the Kaohsiung Medical University Hospital in Taiwan. We excluded subjects with rheumatoid arthritis, cancer, and chronic kidney disease, those <20 or >80 years old, those who failed to offer specimens, those with eGFR <60 mL/min/1.73 m^2^, and those with unfinished questionnaires or incomplete data for covariates. The flowchart for subjects is displayed in Figure 1. This protocol was approved by the Ethics Committee of Kaohsiung Medical University Hospital (approval number: KMUHIRB-G(I)-20210005; date of approval, 19 February 2021). Each participant provided their informed consent before enrollment. The questionnaire collected information on the following demographic characteristics: gender, age, body mass index (BMI), and lifestyle habits (cigarette smoking, alcohol consumption). Exclusively subjects with whole variables (serum creatinine, white blood counts, seven metals and TNF-α concentrations in serum, and renal function) were included in this study. Finally, 421 participants remained in our database. In our study, the outcome variables were white blood counts, TNF-α levels, serum creatinine levels, estimated glomerular filtration rate (eGFR), and the ratio of TNF-α and white blood cells. All outcome variables were log transformed. The measurement of eGFR was based on the Chronic Kidney Disease Epidemiology Collaboration (CKD-EPI) equation, eGFR = 144 × (serum creatinine/0.7)^k^ × (0.993)^age^ [38]. R Statistical Software Version 4.0.3 was used for analysis. GAM and WQS regression used separately the mgcv [39], gWQS [40], and groupWQS [41] packages.

### 2.2. Biomarkers of Multiple Metals and Inflammation

We collected blood samples at annual health examinations and stored them at 4 °C. We used dilution solution, blending solution with 0.2% ammonia solution, 0.1% Triton X-100, and 0.3% HNO3. All blood samples were analyzed via inductively coupled plasma mass spectrometry (ICPMS) of Thermo Scientific XSERIES 2 in the Kaohsiung Medical University Hospital. We tested the following metal and metalloid concentrations in blood: As, Cd, Pb, Se, Co, Cu, and Zn. The procedure for analyses was customized from the method of Ebba Barany in 2007 [42]. We used two standard reference materials (SEMs), ClinChek^®^ RECIPE (Munich, Germany) control Level I No. 8880 and Level II No. 8881, to ensure quality control and quality accuracy during analysis. With the SEMs, coefficients of variation were less than 3% for the measurements at high levels (Level II No. 8881) and were less than 5% for the measurements at low levels (Level I No. 8880). The serum from each subject was separated from coagulated blood samples. Serum concentrations of TNF-α were measured by the protocol of Human TNF-alpha Quantikine ELISA Kit measurement (R&D Systems).

The white blood count and serum creatinine levels were analyzed in the clinical biochemical laboratory in the Kaohsiung Medical University Hospital. All laboratories in the University Hospital were qualified by the Taiwan Accreditation Foundation, proving their quality assurance and quality control (QA/QC).

### 2.3. Statistical Analysis

Characteristics of participants were presented as number (%) or mean ± standard deviation (SD). Pearson correlations were followed to demonstrate the association between metals and metalloids levels and TNF-α. Multiple linear regression models were used to examine the association between serum TNF-α level and the seven elements. Model 2 was adjusted covariates for gender, age, BMI, alcohol consumption, and smoking status. The equation of the multivariable linear regression model was:Y = β_0_ + β_1_ [Metals or Metalloids] + β_2_ [Covariates] + ε(1)

We implemented GAM to survey the dose–response assessment between metals/metalloids and the ratio of TNF-α and white blood cells, adjusted for gender, age, BMI, alcohol consumption, and smoking status. In GAM analysis, the value of the effective degree of freedom (EDF) was greater than 1, which means a complicated relationship between response and predictor. We used smooth functions to predict and plot the relationships between metals/metalloids and the ratio of TNF-α and white blood cells in multivariable GAM.

To identify the association of seven elements and explore the weights of the individual elements, we applied WQS regression. The typical WQS model is as follows:g(u) = β_0_ + β_1_ WQS + z′φ(2)
(3)$$\mathrm{WQS}=\overset{c}{\underset{i-1}{\sum }}w_{i}q_{i}$$

The WQS index represents the weight of each element component. We utilized gWQS regression to certify the association of the mixture with TNF-α, WBC, serum creatinine, the ratio of TNF-α and white blood cells, and eGFR. Estimating the weighted index of the components (e.g., scored into deciles or quartiles) in bootstrap samples of the random training subset and assessing the *p*-value of the weighted index by a holdout dataset in a generalized linear model are crucial parts of gWQS regression. To reduce collinearity and dimensionality, the weights of components are controlled to be 0 to 1. We can identify the prime contribution to the associations between mixtures of levels of As, Cd, Pb, Co, Cu Se, Zn, and TNF-α in serum.

We also employed the grouped WQS regression to identify the health effects of different mixtures. As, Cd, and Pb are known nephrotoxicants [43]. In this model, all elements were divided into toxic metals (As, Cd, Pb) and essential metals (Co, Cu, Se, Zn). We summed up the results using odds ratios and 95% confidence intervals for each element or mixture and evaluated the weights of metal/metalloid levels in the WQS model.

## 3. Results

The demographic characteristics of the study are shown in Table 1. Of all 421 subjects, there were 333 men (79.1%), 129 smokers (30.6%), and 7 alcohol drinkers (1.7%). The mean age was 39.8 years old. The means for the blood elements, TNF-α levels, and eGFR are also presented in Table 1. Pearson correlation coefficients, which measured the relationship between each element and TNF-α levels, were from −0.05 to 0.68, as shown in Figure 2. The highest correlation coefficient is 0.68 (*p* < 0.001), between Pb and TNF-α levels.

### 3.1. Association between Blood Metals/Metalloids and TNF-α

Adjusted for covariates, an increased risk of TNF-α was significantly associated with As (β = 0.030, 95% CI: 0.020, 0.040, *p* < 0.001), Cd (β = 0.161, 95% CI: 0.068, 0.255, *p* < 0.001), Pb (β = 0.005, 95% CI: 0.004, 0.005, *p* < 0.001), Co (β = 0.294, 95% CI: 0.078, 0.509, *p* = 0.007), Cu (β = 0.0005, 95% CI: 0.0002, 0.0008, *p* = 0.003), and Zn (β = 9 × 10^−5^, 95% CI: 6 × 10^−5^, 1.1 × 10^−4^, *p* < 0.001) levels in serum by multivariable linear regression. There were no significant associations with Se. Similarly, the associations for serum creatinine, eGFR, and each serum metal level were significant. Moreover, the associations between Cd (β = 0.038, 95% CI: 0.001, 0.075, *p* = 0.043), Pb (β = 0.0006, 95% CI: 0.0003, 0.0009, *p* < 0.001), Se (β = −0.0008, 95% CI: −0.0012, −0.0003, *p* = 0.001), Cu (β = 0.0002, 95% CI: 0.0001, 0.0004, *p* < 0.001), and Zn (β = 2 × 10^−5^, 95% CI: 5 × 10^−6^, 2.7 × 10^−5^, *p* = 0.004) levels and WBC were significant. There were no significant associations with As and Co. The associations between metal/metalloid levels and TNF-α, WBC, serum creatinine, eGFR, and the ratio of TNF-α and WBC are shown in Appendix A.

### 3.2. GAM Model to Explore the Association between Blood Metals/Metalloids and the Ratio of TNF-α and WBC

Plots for the predicted smooth relationships of each metal/metalloid and the ratio of TNF-α and WBC are presented in Figure 3. The plots show a significant non-linear association between Pb (EDF = 6.28, *p* < 0.001), As (EDF = 2.29, *p* < 0.001), and the ratio of TNF-α and WBC, respectively. On the other hand, the evaluation of metalloids (Co, (EDF = 3.39, *p* = 0.023), Se (EDF = 4.45, *p* = 0.038), Zn (EDF = 1.69, *p* = 0.001)) and the ratio of TNF-α and WBC indicated significant non-linear relationships.

Plots for the predicted smooth relationships of each metal/metalloid and eGFR are illustrated in Figure 3. The plots show a significant non-linear association between Pb (EDF = 3.02, *p* < 0.001), As (EDF = 2.39, *p* < 0.001), Cd (EDF = 4.29, *p* = 0.003) and eGFR, respectively. On the other hand, the evaluation of the metalloids (Co, (EDF = 7.82, *p* < 0.001), Zn (EDF = 3.32, *p* < 0.001), and Cu (EDF = 2.99, *p* < 0.001)) and the eGFR indicated significant non-linear relationships.

### 3.3. WQS Regression to Examine the Association between Blood Metals/Metalloids and TNF-α

In WQS regression analysis, our results showed that the WQS indices were positively associated with TNF-α. We observed that the toxic metal mixtures had significant effects on TNF-α levels (β = 0.314, 95% CI: 0.241, 0.387, *p* < 0.001) after adjusting for gender, age, BMI, alcohol consumption, and smoking status, as shown in Table 2, Appendix A. The highest weighted blood toxic metal in this model was Pb (weighted 63%), followed by As (37%).

We found a similar result for the essential metals model (β = 0.217, 95% CI: 0.140, 0.298, *p* < 0.001), where Zn was highest (63%), followed by Co (26%) and Cu (7%), as shown in Table 2, Appendix A. Within all element mixture models, per quartile increases in serum mixture were associated with TNF-α level (β = 0.351, 95% CI: 0.270, 0.433, *p* < 0.001) (Appendix A).

Pb, As, and Zn mostly contributed to serum TNF-α levels, with 50%, 31%, and 15%, respectively, as shown in Table 3. Per quartile increases in serum mixture were associated with the ratio of TNF-α level and WBC (β = 0.287, 95% CI: 0.199, 0.373, *p* < 0.001) (Appendix A). Pb, As, and Se contributed to the ratio of TNF-α levels with 48%, 41%, and 10%, respectively.

### 3.4. WQS Regression to Examine the Association between Blood Metals/Metalloids and Health Outcomes

The WQS regression models for toxic metal and metalloid mixture indices both were associated with serum creatinine and eGFR but not WBC. We observed that toxic metals mixtures were significantly associated with serum creatinine (β = 0.094, 95% CI: 0.070, 0.118, *p* < 0.001) after adjusting covariates, as shown in Table 2, Appendix A. The weighted serum toxic metals in this association were Pb (87%), As (10%), and then Cd (3%) for serum creatinine. Similarly, eGFR was negatively associated with the toxic mixtures (β = −0.087, 95% CI: −0.108, −0.066, *p* < 0.001) (Appendix A), primarily owing to Pb (90%) and As (8%). As can be seen in Table 2, Appendix A, this model’s indices (β = 0.099, 95% CI: 0.075, 0.123, *p* < 0.001) were associated with serum creatinine due to Zn (79%) and Co (21%). Our results showed the essential metals mixture consisting of Zn (77%) and Co (23%) was significantly associated with eGFR (β = −0.093, 95% CI: −0.115, −0.72, *p* < 0.001) (Appendix A). Within all element mixture models, per quartile increases in blood mixture were associated with eGFR (β = −0.115, 95% CI: −0.138, −0.092, *p* < 0.001), as shown in Table 4 and Appendix A. Zn, Pb, and Co mostly contributed to serum eGFR, with 61%, 24%, and 11%, respectively.

### 3.5. Group WQS Regression to Examine the Association between Blood Metals/Metalloids and TNF-α, and Health Outcomes

Table 4 provides the results of two mixtures, toxic metals and metalloids, with TNF-α levels and other health outcomes through grouped WQS regression after adjusting for age, gender, BMI, smoking status and alcohol consumption status. The mixture of toxic metals was positively significantly associated with TNF-α, WBC, serum creatinine, and the ratio of TNF-α and WBC; the exception was for eGFR (β = −0.066, 95% CI: −0.084, −0.048, *p* < 0.001). The mixture of metalloids was positively significantly associated with WBC and serum creatinine, with the exception of TNF-α (β = −0.091, 95% CI: −0.153, −0.030, *p* = 0.004) and the ratio of TNF-α and WBC (β = −0.079, 95% CI: −0.146, −0.011, *p* = 0.027), eGFR (β = −0.046, 95% CI: −0.061, −0.030, *p* < 0.001).

## 4. Discussion

In our study, we found an association between mixtures of metals/metalloids, TNF-α, and renal function. We utilized different statistical strategies to examine the effects of serum metals and their mixtures. Based on the viewpoint of each element, the multivariable linear regression examined single metal/metalloids and adjusted covariates. This study also found that serum Pb, Se, and Co levels expressed a non-linear relationship with TNF-α and eGFR. Our main finding was the significant relationship between the weights of all types of mixtures and TNF-α and eGFR among workers in metal industries. In addition, the grouped WQS regressions provided quantifiable insight into the individual weights of the mixtures between essential metals and toxic metals. In grouped WQS regressions, the toxic metals significantly increased the level of TNF-α compared with the essential metals.

Whether in the occupational area or general environment, an individual may possibly be exposed to multiple metals everywhere simultaneously [44,45]. Hengstler et al. showed the interaction between the odds of a high level of DNA single-strand breaks (DNA-SSB) and co-exposure to metals, including lead, cobalt, and cadmium. [46]. In addition, early studies reported interactions between essential metals and toxic metals [47]. There is some support in the literature for the current findings, especially regarding the exposure to multiple heavy metals [48,49]. Karri et al. reported significant differences in the expression of protein information between heavy metal mixtures and single metals and demonstrated that the health risks of metal mixture exposure are a greater threat than those of single-metal exposure [50]. Among toxic and essential metals, Turksoy et al. found that the levels of Se and Zn affected the concentrations of TNF-α, IL-6, and IL-10 [51]. Even excessive levels of essential elements may induce adverse health effects. We observed this relationship between Se and Co with TNF-α in the GAM model. Intriguingly, we only observed a significant association between age, TNF-α level, and the ratio of TNF-α level and WBC in the essential metal mixture models.

We observed a relationship between co-exposure to the toxic metal mixture (Pb, As, and Cd) as measured in serum and lower eGFR. Sanders et al. found associations between metals (Hg, As, Cd, and Pb) in urine with eGFR, but there was no significant difference between blood metal mixtures in the general population by WQS regression [52]. Research showed high levels of blood Pb and Cd were associated with renal function (adjusted odds ratios: 2.369, 95%: 1.868–3.004) in a general population of adults from the National Health and Nutrition Examine Survey (NHANES) dataset [53]. Our findings on the toxic mixture model are similar to those identified by Luo et al., who reported Pb was the major association with eGFR compared to the other metals in the mixture model [54]. The protective mechanisms of Zn modifying the relationship between metallothionein and kidney function in vivo have been reported [55], but we did not find similar associations between either the essential metal mixture or any of the other metal mixtures. It is known that the kidney is an organ susceptible to the toxicity of metals due to its ability to reabsorb and condense metals. The degree of harm relies upon the nature, portion, course, and length of openness which might be increased when openness happens across various metals [56]. Tsaih et al. found that longitudinal decreases in renal capacity among middle-aged and older people seem to rely upon both long-term lead exposure and circulating lead, with an impact that is generally articulated among those who were diagnosed with metabolic syndrome, i.e., subjects who were probably a susceptible population [57]. In the three mixture models, we observed a significant association between serum creatinine and gender as well as eGFR and age.

These findings for single metals were in line with previous studies. The levels of TNF-α were affected by occupational exposure of workers to Pb and Cd (median serum TNF-α: 56 pg/mL) versus workers who were not exposed (median serum TNF-α: 22.4 pg/mL) [58,59]. The associations between biomarkers and TNF-α in the study were consistent with other studies which have examined TNF-α levels following exposure to metal/metalloids [10,11]. As exposure increased significant inflammation (serum TNF-α: 5.21 pg/mL) in workers compared to the control group, but Se was associated with lower TNF-α levels in workers [10]. We observed an association between combined exposure to Pb, As, and Zn, as measured in serum, and the level of TNF-α. In our linear regression of single metals, serum As, Cd, Pb, Co, Cu, and Zn levels were significantly associated with TNF-α, but there were no associations with Se. These findings emphasize the possibility of unique health impacts from multiple metal/metalloid co-exposures that would otherwise go undetected in single-metal investigations as well as the efficacy of WQS regression in detecting complicated relationships. We further used the ratio of TNF-α and WBC to model the inflammatory response by WQS regression, in which the component of Se was higher than Zn compared to the outcome of TNF-α level only. The level of Se has the property of alleviating Pb harm through diminishing oxidative pressure via the nucleotide-binding domain, leucine-rich-containing family, and pyrin-domain-containing-3 (NLRP3) inflammasome, according to an animal study [60]. In a cross-sectional study, the authors recommend a decrement in the synthesis of TNF-α; however, our results show the inverse [61]. Future research should examine the contribution of multiple metal/metalloid exposure to immune cell counts (lymphocytes, monocytes, neutrophils) and other health effects among different populations.

A strength of our study was that the WQS regressions and grouped WQS regressions were used to examine the effect of metal/metalloid mixtures on inflammation and renal function in workers. In addition, seven metal/metalloids and TNF-α levels in serum were surveyed to comparatively and comprehensively express internal exposure. We acknowledge additional study limitations. We cannot determine causal relationships between the levels of serum for multiple elements and inflammatory biomarkers, serum creatinine, and renal function in this cross-sectional study. The lack of information on seafood consumption and supplements could lead to a gap in our results [62]; however, compared to occupational exposure, the effect of seafood was limited. On the other hand, we used metals in the blood as biomarkers that included every source of metals, including food, seafood, and water. In addition to these limitations, serum or urine cotinine may be a better biomarker for assessing smoking exposure, rather than only a questionnaire [63]. Another limitation was that we did not collect urine samples from workers; therefore, we could not investigate the species forms of As in urine, such as dimethylarsinic acid, monomethylarsonic acid, arsenocholine, and arsenobetaine.

## 5. Conclusions

This study demonstrated that metal/metalloid mixtures in serum were associated with TNF-α levels and eGFR by GAM. Heavy metals are still one of the crucial environmental hazards associated with inflammation and kidney health. The effects of metal co-exposure are likely modified by the concentration and duration of exposure, genetic predisposition, and other individual factors. Further epidemiological studies in a framework with more inflammatory cytokines and biomarkers of oxidative stress and controlled environmental and individual factors would increase the clarity regarding the mechanisms of the metal/metalloid mixtures in occupational exposure.

## Figures and Tables

**Figure 1 ijerph-19-07399-f001:**
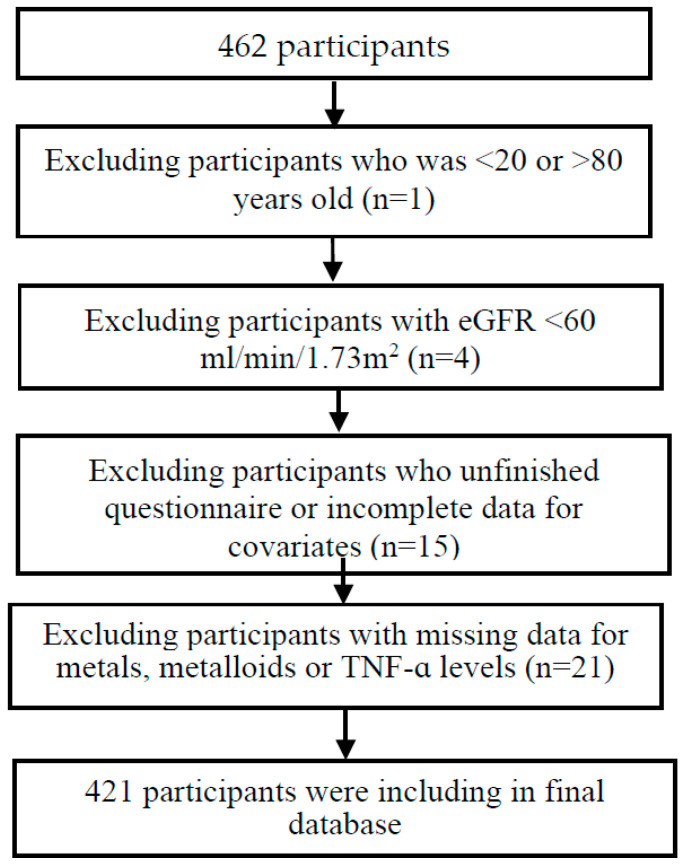
The number of the participants changes in the study.

**Figure 2 ijerph-19-07399-f002:**
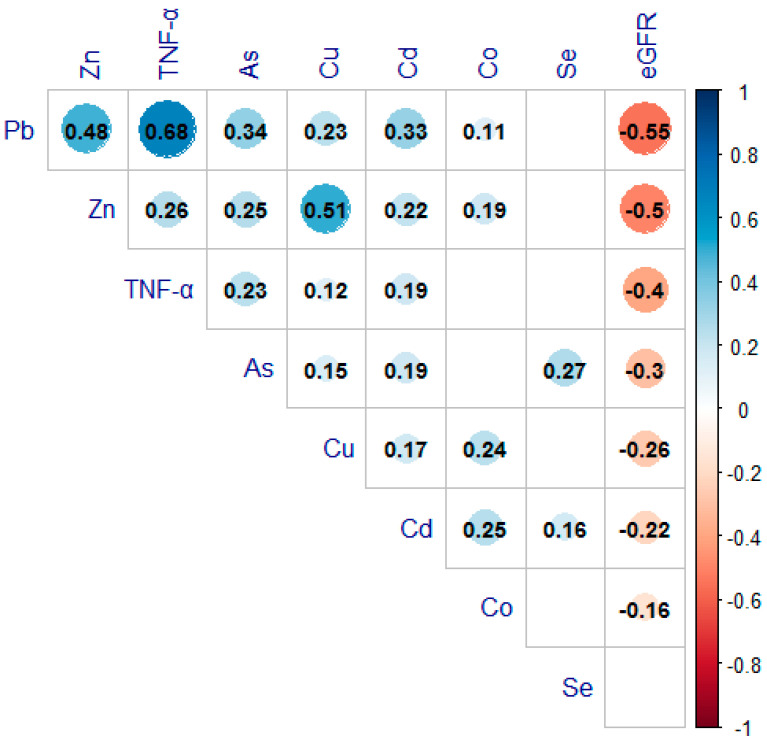
The Pearson correlation analysis between multiple metal concentrations and TNF-α level, and eGFR. (If the *p*-value was non-significant, the correlation coefficient is not be displayed).

**Figure 3 ijerph-19-07399-f003:**
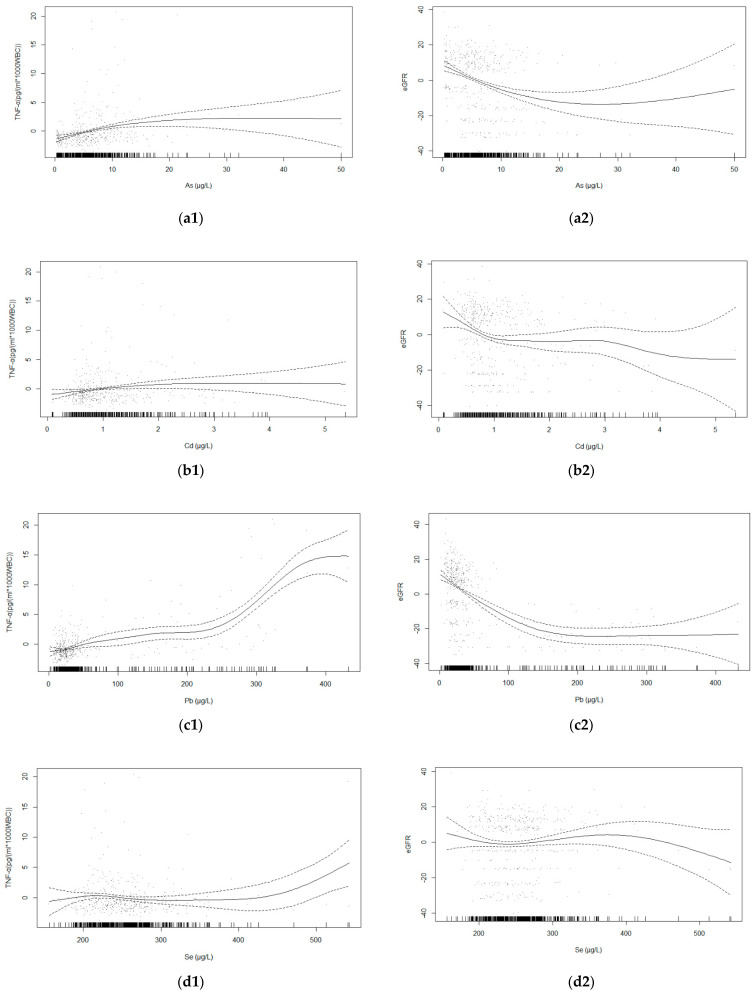
Relationships between (**a1**) As (*p* < 0.001), (**b1**) Cd (*p* = 0.062), (**c1**) Pb (*p* < 0.001), (**d1**) Se (*p* = 0.037), (**e1**) Co (*p* = 0.023), (**f1**) Cu (*p* = 0.165), (**g1**) Zn (*p* < 0.001) concentrations and the ratio of TNF-α and WBC based on generalized additive model after adjusting for age, gender, BMI, smoking status and alcohol consumption status. In addition are the relationships between (**a2**) As (*p* < 0.001), (**b2**) Cd (*p* = 0.003), (**c2**) Pb (*p* < 0.001), (**d2**) Se (*p* = 0.255), (**e2**) Co (*p* < 0.001), (**f2**) Cu (*p* < 0.001), (**g2**) Zn (*p* < 0.001) concentrations and eGFR based on adjusted generalized additive model.

**Table 1 ijerph-19-07399-t001:** General characteristics of the study population (N = 421).

Characteristics	Mean ± SD	Min–Max
Gender		
Male	333 (79.1%)	
Female	88 (20.9%)	
Cigarette smoking		
Yes	129 (30.6%)	
No	292 (69.4%)	
Alcohol consumption		
Yes	7 (1.7%)	
No	414 (98.3%)	
Age (years)	39.8 ± 8.2	23.0–79.9
BMI (kg/m^2^)	24.8 ± 3.5	17.1–35.0
White blood cells (10^3^/µL)	6.8 ± 1.6	2.7–13.2
Serum creatinine (mg/dL)	0.9 ± 0.2	0.3–1.3
TNF-α (pg/mL)	23.8 ± 22.6	7.8–170.8
Arsenic (μg/L)	6.4 ± 5.2	0.2–50.0
Cadmium (μg/L)	1.1 ± 0.7	0.1–5.4
Lead (μg/L)	56.0 ± 76.4	1.9–432.0
Selenium (μg/L)	255.6 ± 49.9	155.9–542.2
Cobalt (μg/L)	0.5 ± 0.3	0.2–2.9
Copper (μg/L)	921.7 ± 171.1	494.2–2224.7
Zinc (μg/L)	7625.3 ± 1992.4	3928.7–21,106.3
eGFR (mL/min/1.73 m^2^)	99.5 ± 17.9	61.8–139.6

**Table 2 ijerph-19-07399-t002:** Results for toxic metals and essential metal mixtures by WQS regression analyses after adjusting for age, gender, BMI, smoking status, and alcohol consumption status.

Toxic Metals				
Outcomes	Estimates (95% CI)	*p*-Value	Metal/Metalloids with Weight	Components Weight
TNF-α	0.314 (0.241, 0.387)	<0.001	Pb, As	63%, 37%
White blood cells	0.023 (−0.010, 0.056)	0.168	n/a	n/a
TNF-α/WBC	0.279 (0.120, 0.358)	<0.001	Pb, As	52%, 48%
Serum creatinine	0.094 (0.070, 0.118)	<0.001	Pb, As, Cd	87%, 10%, 3%
eGFR	−0.087 (−0.108, −0.066)	<0.001	Pb, As, Cd	90%, 8%, 2%
**Essential Metals**				
**Outcomes**	**Estimates** **(95% CI)**	***p*-Value**	**Metal/Metalloids with Weight**	**Components Weight**
TNF-α	0.217 (0.136, 0.298)	<0.001	Zn, Co, Cu, Se	63%, 26%, 7%, 4%
White blood cells (WBC)	0.019 (−0.010, 0.049)	0.201	n/a	n/a
TNF-α/WBC	0.194 (0.084, 0.304)	0.001	Zn, Se, Co, Cu	32%, 29%, 28%, 11%
Serum creatinine	0.099 (0.075, 0.123)	<0.001	Zn, Co	79%, 21%
eGFR	−0.093 (−0.115, −0.072)	<0.001	Zn, Co	77%, 23%

**Table 3 ijerph-19-07399-t003:** Results of all metal mixtures by WQS regression analyses.

Outcome	Model 1				Model 2 *			
	Estimates (95% CI)	*p*-Value	Metal/Metalloids with Weight	Components Weight	Estimates (95% CI)	*p*-Value	Metal/Metalloids with Weight	Components Weight
TNF-α	0.368 (0.289, 0.448)	<0.001	Pb, As, Zn	42%, 37%, 15%	0.352 (0.270, 0.433)	<0.001	Pb, As, Zn	50%, 31%, 15%
WBC	0.034 (−0.001, 0.069)	0.062	n/a	n/a	0.018 (−0.017, 0.053)	0.307	n/a	n/a
TNF-α/WBC	0.318 (0.231, 0.406)	<0.001	As, Pb, Co	51%, 30%, 15%	0.287 (0.199, 0.373)	<0.001	Pb, As, Se	48%, 41%, 10%
Creatinine	0.143 (0.116, 0.171)	<0.001	Zn, Pb, As	55%, 42%, 7%	0.124 (0.098, 0.150)	<0.001	Zn, Pb, Co	61%, 25%, 9%
eGFR	−0.122 (−0.146, −0.098)	<0.001	Zn, Pb, Co	56%, 30%, 12%	−0.115 (−0.138, −0.092)	<0.001	Zn, Pb, Co	61%, 24%, 11%

* After adjusting for age, gender, BMI, smoking status, and alcohol consumption status.

**Table 4 ijerph-19-07399-t004:** Results of adjusted group WQS regression analyses.

Outcome	Essential Metals Estimates (95% CI)	*p*-Value	Toxic Metals Estimates (95% CI)	*p*-Value
TNF-α	−0.091 (−0.153, −0.030)	0.004	0.270 (0.214, 0.326)	<0.001
White blood cells	0.030 (0.005, 0.054)	0.018	0.026 (0.004, 0.048)	0.019
TNF-α/WBC	−0.079 (−0.146, −0.011)	0.027	0.257 (0.198, 0.316)	<0.001
Serum creatinine	0.086 (0.066, 0.105)	<0.001	0.098 (0.077, 0.119)	<0.001
eGFR	−0.046 (−0.061, −0.030)	<0.001	−0.066 (−0.084, −0.048)	<0.001

## Data Availability

The data could be applied to use via an application proceeding at Kaohsiung Medical University.

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
