# Peer review of "Use of Generalized Weighted Quantile Sum Regressions of Tumor Necrosis Factor Alpha and Kidney Function to Explore Joint Effects of Multiple Metals in Blood"

_ijerph, 2022, doi:10.3390/ijerph19127399_

Round 1

Reviewer 1 Report

The manuscript entitled “Use of Generalized Weighted Quantile Sum Regressions of Tumor Necrosis Factor Alpha and Kidney Function to Explore Joint Effects of Multiple Metals in Blood” aims to explore the association between heavy metals and essential elements in blood and TNF-α as well as kidney function in 421 workers. The obtained results suggested that Pb, As, Zn, Se, and their mixtures may act on TNF-α through interactive mechanisms offering insights into what primary components of metal mixture affect inflammation and kidney function during co-exposed to metals.

Authors are mentioning occupational exposure to heavy metals but be interesting to mention how different dietary preferences can impact the body load of heavy metals and essential elements as well.

Page 11, line 269 – “Hengstler et al. showed that the severity of DNA damage effect by multiple metals more than resulting in single toxic metal” – please rephrase this sentence to make it clearer.

Minor remarks:

Page 2, line 61 – change to “Studies using in vitro models with Cd, Pb, and As showed…”

Page 2, line 63 – change to “inflammatory markers [23,24] and blood cadmium level…”

Page 2, line 72 – change to “therefore the health effect of metal co-exposure should have more attention”

Reviewer 2 Report

The manuscript entitled “Use of Generalized Weighted Quantile Sum Regressions of Tumor Necrosis Factor Alpha and Kidney Function to Explore Joint Effects of Multiple Metals in Blood” aimed to evaluate the association between heavy metals exposure and the blood level of TNFα and kidney function. The mixture of Pb, As and Zn was positively associated with TNFα and kidney function. The manuscript is well written, but there are some aspects that should be clarified before acceptance:

1.      Please indicate the methods used and reagents for the determination of white blood counts, serum creatinine

2.      In the material and methods please indicate what software you used for statistical analysis

3.      A graphical abstract can be helpful for the reader. 

Reviewer 3 Report

-In figure 1, please indicate the number of participants excluded due to unfinished questionnaire. 

-Please provide the rationale of selecting these metal elements. 

-Figure 3 was not well explained and the quality was low.

-In the discussion, please indicate that were there any sex- or age-dependent patterns. 

Reviewer 4 Report

Review on the manuscript of Luo KH. et al.: “Use of Generalized Weighted Quantile Sum Regressions of Tumor Necrosis Factor Alpha and Kidney Function to Explore Joint Effects of Multiple Metals in Blood”.

This manuscript explores a putative relationship between the blood levels of metals/metalloids and TNF-α levels or renal function. The authors found a significant correlation between the blood levels of As, Cd, Pb, Co, Cu and Zn and the levels of TNF-α, the serum creatinine, and the eGFR. Using the GAM model, the authors show: (1) a significant non-linear association of Pb and TNF-α levels and of As and white blood cells (WBC); Co, Se, Zn and the ratio of TNF-α and WBC showed non-significant relationship; (3) a significant non-linear association of Pb, As and Cd with the estimated glomerular filtration rate (eGFR); (4) a non-significant relationship between Co, Zn or Cu and the eGFR. Using the WQS regression, the authors show: (1) significant effects of mixtures of toxic metals on TNF-α levels, serum creatinine and eGFR; (2) a positive significantly association between the mixture of toxic metals and the TNF-α, WBC, serum creatinine and the ratio of TNF-α and WBC; (3) a positive significantly association between the mixture of metalloids and the WBC and serum creatinine levels.

The data shown in the manuscript seem to be clear. However, some issues that arise, which need to be clarified in more detail, are listed below for consideration of the authors.

1 – Several sentences in the manuscript make no sense. For example: “A previous study of Zn possibly had protective mechanisms with metallothionein to kidney function”. In general, I think I could get the authors’ idea, but in other cases it is difficult to find a logic on it. Thus, I would recommend the authors to have their work proofread by an English native speaker. It would help the overall quality of the work.

2 – While describing the results, I felt that the information could be more systematized and presented in a clearer way. In my opinion, for most of the cases the tables are enough, with no need of detailed description of the data (sometimes the description makes the interpretation very hard).

3 – As indicated by the authors in the discussion section, there are reports in the literature on the same topic. Based on this, what is the novelty of this study? Does it add substantial knowledge on this topic?

4 – As stated by the authors, “the lack of information on seafood consumption and supplements could lead to a gap in our results”. Selenium and zinc are in fact two elements present at high levels in a great variety of supplements. Considering the lack of information on the consumption of supplements by the study participants, it is difficult to take valid conclusions out of it.

5 – I would recommend the authors to indicate the software they used for the statistical calculations.

Round 2

Reviewer 2 Report

The authors addressed my comments. The manuscript is much improved compared with the first version and can be accepted. 

Reviewer 4 Report

Second review on the manuscript of Luo KH. et al.: “Use of Generalized Weighted Quantile Sum Regressions of Tumor Necrosis Factor Alpha and Kidney Function to Explore Joint Effects of Multiple Metals in Blood”.

This manuscript explores a putative relationship between the blood levels of metals/metalloids and TNF-α levels or renal function. The authors found a significant correlation between the blood levels of As, Cd, Pb, Co, Cu and Zn and the levels of TNF-α, the serum creatinine, and the eGFR. Using the GAM model, the authors show: (1) a significant non-linear association of Pb and TNF-α levels and of As and white blood cells (WBC); Co, Se, Zn and the ratio of TNF-α and WBC showed non-significant relationship; (3) a significant non-linear association of Pb, As and Cd with the estimated glomerular filtration rate (eGFR); (4) a non-significant relationship between Co, Zn or Cu and the eGFR. Using the WQS regression, the authors show: (1) significant effects of mixtures of toxic metals on TNF-α levels, serum creatinine and eGFR; (2) a positive significantly association between the mixture of toxic metals and the TNF-α, WBC, serum creatinine and the ratio of TNF-α and WBC; (3) a positive significantly association between the mixture of metalloids and the WBC and serum creatinine levels.

In the revised version of the manuscript, the issues raised previously were clarifyed. So, I consider that, now, the overall quality of the manuscript is improved.